# Tailored Approach and Multimodal Intraoperative Neuromonitoring in Cerebellopontine Angle Surgery

**DOI:** 10.3390/brainsci12091167

**Published:** 2022-08-31

**Authors:** Alessandro Izzo, Vito Stifano, Giuseppe Maria Della Pepa, Michele Di Domenico, Quintino Giorgio D’Alessandris, Grazia Menna, Manuela D’Ercole, Liverana Lauretti, Alessandro Olivi, Nicola Montano

**Affiliations:** 1Department of Neurosurgery, Fondazione Policlinico Universitario A. Gemelli IRCCS, Università Cattolica del Sacro Cuore, Largo A. Gemelli, 8, 00168 Rome, Italy; 2Department of Neuroscience, Neurosurgery Section, Università Cattolica del Sacro Cuore, Largo F. Vito, 1, 00168 Rome, Italy

**Keywords:** cerebellopontine angle, CPA, microsurgery, neuronavigation, IONM, neuromonitoring

## Abstract

The cerebellopontine angle (CPA) is a highly complex anatomical compartment consisting of numerous nervous and vascular structures that present mutual and intricate spatial relationships. CPA surgery represents, therefore, a constant challenge for neurosurgeons. Over the years, neurosurgeons have developed and refined several solutions with the aim of maximizing the surgical treatment effects while minimizing the invasiveness and risks for the patient. In this paper, we present our integrated approach to CPA surgery, describing its advantages in treating pathologies in this anatomical district. Our approach incorporates the use of technology, such as neuronavigation, along with advanced and multimodal intraoperative neuromonitoring (IONM) techniques, with the final goal of making this surgery safe and effective.

## 1. Introduction

The cerebellopontine angle (CPA) is probably the most fascinating compartment in neurosurgery due to the technical challenges that this surgery implies [1]. The anatomy related to the retrosigmoid approach and the CPA has been extensively described in many papers and textbooks in the past years. It is worth mentioning that in order to simplify the CPA approach, Rhoton proposed the “anatomical rule of three”, dividing the CPA into three compartments according to the three pairs of cerebellar arteries arising from the vertebrobasilar system [2]. He identified: (1) the “upper neurovascular complex”, following the superior cerebellar artery and including the midbrain and upper pons, trigeminal nerve, trochlear nerve, superior petrosal vein and its tributaries, and superior cerebellum; (2) the “middle neurovascular complex”, following the anterior inferior cerebellar artery and including the middle pons, middle cerebellar peduncle, abducens, and facial and vestibulocochlear nerves; and (3) the “lower neurovascular complex”, following the posterior inferior cerebellar artery and including the medulla, lower cranial nerves, and inferior cerebellum. Over the past years, the retrosigmoid craniotomy has evolved, initially from a bilateral suboccipital to a unilateral approach. Cushing reported a very impressive morbidity and mortality rate (about 80%) using the unilateral suboccipital approach and proposed a large bilateral suboccipital craniotomy in order to rapidly open the cisterna magna and obtain CSF release. Nonetheless, Dandy continued to promote the unilateral approach, combining CSF release, cerebellar retraction, and, eventually, amputation of a portion of the cerebellum to gain access to the CPA. In recent years, the use of surgical microscopes has enabled the visualization of the critical neural and vascular structures, significantly increasing the possibility of a CPA tumor radical resection with cranial nerve function preservation. A CPA approach is needed for a variety of diseases, mainly extra-axial neoplasms such as vestibular schwannomas and meningiomas, and neurovascular conflicts. Plenty of nervous and vascular structures are packed in the CPA, and all of them must be carefully preserved in order not to impair patients’ well-being. To this aim, several tools have been developed to help neurosurgeons in maximizing the cure rate while preserving neurological functions. In the following paragraphs, we will describe our tailored and multimodal approach to CPA surgery, highlighting the indications and usefulness of the various techniques and detailing useful tips and tricks for the neurosurgery operating room.

## 2. Operating Room Tools

### 2.1. Neuronavigation

In CPA surgery, localizing the anatomical structures before craniotomy is of paramount importance. Adequate exposure of such structures allows an optimal surgical corridor while minimizing cerebellar retraction and manipulation. The most common surgical approach used to treat CPA diseases, namely, the retrosigmoid approach, aims to achieve an extracerebellar surgical corridor to reach the CPA. In this regard, placing the burr-hole at the transverse-sigmoid junction (TSSJ) enables an appropriate, prompt, and safe exposure of the sinuses. This brings several benefits, including less cerebellar retraction, a reduction of bone defects that would result from extensive bone drilling in the case of a burr-hole too far from the TSSJ, less operative time, and a decrease in risk of sinus injury and postoperative CSF leaks [3,4,5]. Traditionally, the localization of the venous sinuses relied on methods that exploit anatomical landmarks and craniometric points. However, these landmarks are sometimes not visible intraoperatively and display interpatient variability; additionally, there is a lack of consensus on which landmarks are the most reliable for localizing the TSSJ [5]. The surgeon’s experience is, therefore, necessary to interpret the patient-specific anatomy on a case basis in order to identify the real position of the relevant anatomical structures and to tailor the craniotomy accordingly. Nowadays, neuronavigation is a common and affordable technology to localize anatomical structures, helping surgical orientation. Several papers have described its implementation during posterior fossa surgery [3,4,6,7,8,9]. In our surgical activity, we routinely use the park-bench position for the retrosigmoid approach and the neuronavigation system during the craniotomy planning for both tumor and functional cases (Figure 1, Figure 2 and Figure 3). In CPA surgery, neuronavigation is especially useful in the identification of TSSJ. In our experience, this makes the venous sinuses’ exposure unnecessary (Figure 3), reducing the possibility of venous sinus injury [1]. Furthermore, this technology allows us to tailor the craniotomy for each patient, guaranteeing an optimal surgical corridor while minimizing the risk of surgical complications and reducing surgical time. In addition, neuronavigation also brings advantages in the educational field, easing the explanation of the complex topographic anatomy of the CPA to trainees.

### 2.2. Intraoperative Neuromonitoring (IONM)

IONM techniques in CPA surgeries are widely utilized with the goal of preserving function and preventing the injury of several neural structures. In fact, during CPA surgical procedures, the brainstem and cranial nerves might be injured, with possible devastating postoperative deficits. The ideal monitoring system should provide continuous and quick feedback to the surgeon and the anesthesiologist, based on reproducible, easily interpretable data. IONM should be able to detect potentially dangerous situations and identify critical neurophysiological variations. The rapid interpretation of these data and prompt communication to the team should be adequate to revise the surgical plan and prevent irreversible neurological injuries. Moreover, the “perfect IONM technique” should indicate a detailed profile of injury localization, gravity, timing, and prognosis for recovery.

In order to obtain a standardized method as close as possible to this hypothetical model, our IONM protocol in CPA surgery includes:Somatosensory evoked potentials (SEPs);Motor evoked potentials (MEPs);Cortico-bulbar motor evoked potentials (cMEPs);Brainstem auditory evoked potentials (BAEPs);Free-running electromyography (EMG);Lateral spread response (LSR);Direct electrical stimulation (DES) of the nerves.

As suggested by the Italian recommendations from the interdisciplinary panel of the Italian Society of Clinical Neurophysiology and the Italian Society of Neurosurgery [10], the technician of our service has the specific responsibility of settingup the necessary equipment and collaborates with the responsible IONM service in applying the protocol and interpreting the data. The person in charge is a physician with appropriate training and intimate knowledge of what is happening clinically and surgically at each step. The presence of a physician in the OR is mandatory during some surgical procedures (as in CPA surgery) when there is a need for monitoring and testing/mapping of neurological functions to gain functional identification of “eloquent” neural structures.

#### 2.2.1. Electrode Placements

After skin cleaning, monopolar needle electrodes are inserted into the target muscles and fixed with tape or Tegaderm™ following the belly-tendon montage technique, with the active electrode inserted within the belly muscle and the reference electrode close to the tendon. The target muscles listed below are selected to register valid corticospinal and cortico-bulbar responses and any electromyographic changes involving cranial nerves.

For upper and lower limbs bilaterally:Biceps brachii;Abductor pollicisbrevis muscles;Tibialis anterior;Quadriceps;Abductor hallucis.

For motor cranial nerves:Orbicularis oris (VII C.N);Orbicularis oculi (VII C.N);Masseter (V C.N);Trapezius (IX C.N);Vocal cords (using The NIM TriVantage^®^ EMG Tube (Medtronic)) (X C.N);Tongue (XII C.N).

Skin disc-adhesive electrodes are placed to stimulate the median nerve in the middle of the volar wrist. Nine corkscrew electrodes (C; C2; C3′; C4′; Cz′; Fz; Fz′; Cz; C3 or C4, depending on the affected site) are placed at the scalp, following the International 10–20 system. As reported [11], the “prime” mark indicates a modified site located 2 cm posterior to the named International 10–20 system scalp site. Figure 4 indicates the correct localization of the electrodes in our practice.

In Table 1, the parameters of stimulation and registration of evoked potentials are reported.

#### 2.2.2. SEPs

SEPs registration is used for monitoring the intracranial lemniscal sensory system as it traverses the brainstem and cerebral hemispheres [12]. For studying the upper extremities, the stimulation is produced at the wrist, with thumb adduction as movement evoked. Stimulus intensity is regulated to cause a 1–2 cm movement. In our experience, we prefer to use superficial disc electrodes, as previously reported. The recording electrodes are placed on the scalp in the C3′ and C4′ positions, while another electrode in the Fz position is used as reference. Stimulated SEPs are recorded with the following parameters: single pulse, 300–500 µs pulse duration, 4.1 Hz, registration: C3′–Fz (right stimulus), C4′–Fz (left stimulus). Stimulation and recording settings should lead to stable, low-noise, easily reproducible potentials. We can identify the N20 by analyzing peak morphology, latency, and amplitude (Figure 5). The N20 is a negative wave observed with a latency of about 20 msec, representing the activation of the primary somatosensory cortex in the hand area. This is a well-defined peak, registered at C3′ or C4′, because of the significant cortical representation of the hand at the level of the sensory areas. As reported, primary measurements are the N20 peak amplitude and latency. Latencies should stay within about 5–10% of the baseline value. Amplitudes should stay within 50% of the baseline. Our monitoring team has considerable experience in detecting possible external factors as the cause of registered variability. Every time a considerable change in amplitude and latency is detected, a rapid assessment is needed to identify the possible systemic, surgical, or technical cause in order to minimize false-positive rates and avoid confounding communication to the surgical staff [13]. A quick summary of possible sources of SEPs variations is reported in Table 2. In our routine practice, a variation of 30% from the baseline value with a suggested surgical cause is always communicated to the surgical staff.

#### 2.2.3. MEPs

MEPs monitoring allows the ongoing assessment of motor tract function during the operative procedure [14]. Transcranial stimulation activates spinal motor pathways selectively, producing compound muscle action potentials (CMAPs). Because of the variability of activation of low-threshold spinal alpha motor neurons by each descending corticospinal volley, sequential transcortical electric stimuli produce CMAPs that could vary significantly in waveform, amplitude, and morphology (Figure 6). The produced CMAPs are generally associated with some movements of the patients, even in the presence of incomplete anesthesiological muscle blockade [15]. MEPs are recorded with the following parameters: C1–C2 (anodal), train of stimuli 5–7 pulses, 500 µs pulse duration, 250–500 Hz. Stimulus intensity required for effective transcranial stimulation varies significantly among patients, and individual adjustment to obtain the best response without extreme muscular activation is necessary. We generally perform MEPs intermittently rather than continuously, especially during critical surgical steps where patient movements are not desirable. No well-defined guidelines exist regarding MEP interpretation. In our practice, we consider a critical reduction in amplitude and wave complexity not otherwise explained a signal of potential damage, even though sometimes the final clinical outcome correlates simply with the presence vs. absence of CMAP responses. Moreover, provided stable anesthesia and neuromuscular blockade are achieved, the increase in voltage threshold needed to induce a CMAP response, unexplained by other factors, is used as an alert of potential motor tract injury [16].

#### 2.2.4. cMEPs

Even though the anatomical details of cortico-bulbar tracts are still poorly known, cMEPs are a relatively new and valuable IONM technique to monitor the functional and anatomical integrity of cranial nerves during CPA surgery. Generally, the chosen cathode position is Cz and the anode is placed in C3 or C4 in order to stimulate the motor area contralateral to the tumor side. Since transcranial electrical stimulation is applied at the scalp over the motor area representing those muscles innervated by cranial motor nerves, the parameters of stimulation to obtain the cMEPs are identical for all cranial motor nerves. As reported by our team for extra-axial CPA tumors [17], the stimulation technique for cMEPs is as follows: single pulse 350–500 µs; train of pulses 4–7 stimuli, 350–500 µs, 500–700 Hz, delivered at 40 ms from the first single pulse (Figure 7). The main point of this technique is registering the response derived by true activation of the multi-synaptic cortico-bulbar pathway and not from direct stimulation of the nerves. This latter event is always possible, especially when high intensity is needed for stimulation. As a practical rule, if a response is induced after a train of pulses, it is interpreted as a “central response”. Otherwise, a response evoked after a single pulse is considered an effect of the direct activation of the nerve and interpreted as a “peripheral response” or a mixture of central and peripheral responses. Moreover, we consider as reliable a potential with at least 80 µV of amplitude, stable and replicable. Our experience in CPA surgery for vestibular schwannoma and other extra-axial tumors outlines the importance of obtaining cMEPs from both orbicularis oculi (OOc) and orbicularis oris (OOr) muscles to have complete facial nerve monitoring during the dissection and resection maneuvers [17]. In particular, the percentage ratio between the final and the baseline facial nerve MEP amplitudes represents a potential tool to predict early and late postoperative facial nerve function [17]. To validate cMEPs as a reliable tool to predict the function of other cranial nerves, similar algorithms have been proposed [18,19]. However, the presence of several branches of cortico-bulbar tracts, with multiple innervations for cranial nerve nuclei, and the possibility of complex redundant circuits could make the analysis of the relationships between the change of cMEP amplitudes or thresholds and the clinical outcome more challenging.

#### 2.2.5. BAEPs

BAEPs allow the study of the integrity of the auditory pathway, monitoring the circuitry of the brainstem and the auditory nerve response. The stimulus is a “click” at high frequency (19–21 Hz) and high intensity (80–100 dB), administered at the external auditory canal by means of specific electrodes. The recording is carried out with electrodes positioned on the ear lobe bilaterally, referring to a cephalic electrode placed in the Cz′ position. The short latency BAEP involves a complex polyphasic wave, with different components numbered from I to V. Analyzing this response, it is possible to evaluate the potentials generated from, respectively, the cochlear nerve (I), the root entry zone (II), the nuclear complex (III), the superior olive (IV), and the lateral lemniscus (V). These responses are registered within 6msec after the stimulus. Because of the modality and number of acquisitions (1000 stimuli in our practice) and the artifacts generated by neural manipulation, the debate concerning the usefulness of intraoperative BAEPs is still ongoing. Although BAEPs reportedly give accurate feedback on brainstem damage, their reliability in predicting hearing preservation is more questionable [20,21]. The presence, amplitude, and latency of waves I, III, and V are candidate predictors. During IONM, our policy is to notify the surgical staff of a 10–20% change in latency and a loss of more than 50% in wave amplitude. It is essential to know that, although the presence of wave V significantly predicts hearing preservation, the absence of wave V does not preclude preserved hearing.

#### 2.2.6. Free-Running EMG

Continuous EMG is used to record evident changes in voluntary motor unit potentials and represents relatively non-invasive real-time information regarding the integrity of motor neurons [22]. Every mechanical, thermal, or metabolic irritation may determine axonal membrane depolarization and induce a recordable potential. The potentials of interest are motor unit potentials and neurotonic discharges. Spikes or burst activity can be observed over the background activity in the case of single or brief sequences of motor unit activation. Neurotonic discharges are defined as a sustained high-frequency EMG activity pattern. These potentials are morphologically heterogeneous and must be distinguished from a variety of other muscle activities, such as fibrillation and spontaneous contraction, as well as from artifactual waveforms. Artifactual activity is primarily related to the electrical devices in the operating rooms and to accidental movements of the needle electrode. These “false activations” could be easily recognized by their irregular form and are especially visible when using cautery or ultrasound aspiration. Nonetheless, the presence of these continuous artifacts could hide pathological signals from the EMG. Finally, neurotonic discharges may occur without mechanical injury, for example, during (cold) water irrigation within the surgical field, especially in the case of stressed or partially damaged nerves. Although several criteria have been proposed to identify EMG activity patterns that are suspicious for nerve injury, the terminology remains confusing, and a strict correlation between EMG activity and clinical outcome is still lacking [23]. EMG activity patterns could relate to mechanical stress in the case of the distraction, compression, or retraction of the nerve or thermal stress due to cauterization, leading the surgeon to adopt the correct maneuvers in order to avoid injuries.

#### 2.2.7. LSR

Aberrant electric activity of the facial nerve can be detected and registered in the case of surgery for hemifacial spasm. This pathological condition is characterized by intermittent contractions of the muscles innervated by the facial nerve due to the vascular compression of the nerve root entry zone. From a neurophysiological point of view, the stimulation of a branch of the facial nerve spreads to muscles innervated by other branches. This phenomenon, known as the LSR, can be continuously monitored during microvascular decompression surgery to confirm the adequacy of surgical maneuvers [24]. Although previous studies have demonstrated a strong correlation between optimal decompression of the entry zone and the intraoperative disappearance of LSR [25], other studies have underlined the role of LSR modification with the correct identification of the vessel responsible for the conflict and the persistence of LSR as predictive of an increased rate of hemifacial spasm recurrence [26].

In our clinical practice, the stimulation is provided from the mandibular and temporal branches of the facial nerves and registered at the OOc, OOr, and mentalis muscles. The stimulus is delivered at low frequency and at an intensity of 3 to 10 mA. The anomalous registration is detected at the level of the OOc in the case of mandibular stimulation and at the level of the mentalis and/or OOr in the case of temporal stimulation.

#### 2.2.8. DES

DES of the cranial nerves can be used to correctly identify the neural structures (especially in the case of distorted anatomy because of the tumor). When possible, DES performed on the portion of the nerve proximal to the tumor can confirm the integrity of the nerve throughout the procedure leading the surgeon to a safer and adaptable operating strategy (Figure 8). This technique is helpful in localizing the nerves when they are not anatomically identifiable, as well as in confirming the function of nerves that are intimately adherent to the tumor capsule during surgical manipulation.

Monopolar constant-current low-frequency stimulation is used for DES. An intensity of stimulus at 0.1 mA, delivered on the nerve surface, is enough to record a CMAP. The need for higher stimulus intensity might indicate a relative distance to the neural structure or the presence of a relevant tissue barrier between the probe and the nerve. The more the motor response is evoked with low intensity, the more the surgeon is working close to the nerve. At the end of the surgery, especially for the facial nerve function, a proximal stimulation at a low threshold (0.1–0.3 mA) seems to correlate with a favorable facial nerve outcome [27]. The absence of an evoked CMAP response from proximal stimulation, despite a preserved response distal to the surgical site, may indicate intraoperative injury, although it is not possible to differentiate neurapraxia from axonal injury acutely [28]. DES is definitely a reliable parameter, but this is not always promptly available, especially in the earliest stages of surgery. cMEPs provide a real-time assessment of neural functional integrity, especially for the facial nerve, during any surgical step, even when neural structures and their course are difficult to identify, as in the case of large lesions or at early stages of the debulking. Moreover, the site of nerve injury may be localized relatively far from the surgical area and cannot be detected by DES. This is particularly true in large tumors when early manipulation and traction during initial debulking can produce injury in the not-exposed segments [17]. For these reasons, we think that the integration of DES with cMEPs is mandatory for this kind of surgery.

## 3. Discussion

A progressive multimodal approach to CPA surgery has evolved in the last decade. The application of neuronavigation has almost become a standard of care in order to design a tailored and safe approach to the posterior fossa, avoiding possible damage to vascular structures and reducing the risk of suboptimal exposure. In CPA surgery, sinus exposure can be a demanding procedure, requiring significant bone drilling and loss. Moreover, the position of the asterion, which is the main bone landmark, corresponds to the TSSJ in only 23% of cases [7]. The use of neuronavigation in CPA surgery allows us to identify the key point for the craniotomy, just behind and below the TSSJ, making visual identification of the TSSJ unnecessary and significantly reducing the risk of venous sinus injury, the craniotomy size, and, finally, surgical time [3]. Nonetheless, despite these technical advances, CPA surgery remains a challenge for neurosurgeons because of the difficulties linked to the brainstem and cranial nerve function preservation. This is especially true in cases of large CPA tumors [29]. In these cases, neural structure identification can be difficult to achieve at the beginning of the operation. Thus, IONM techniques are needed in CPA surgery to assess the anatomical and functional integrity of neural structures by mapping and testing specific neural pathways. Obviously, due to the costs of these tools, it is impossible to use all these techniques for every case. For example, in cases of microvascular decompression of the trigeminal nerve, we routinely use neuronavigation to plan the craniotomy, but IONM is not mandatory due to the fact that the anatomy is usually preserved. On the other hand, we consider IONM mandatory in cases of CPA tumors to preserve the brainstem and cranial nerve function [17,29]. Furthermore, IONM can be very useful in cases of microvascular decompression of the VII cranial nerve for hemifacial spasm in order to study the LSR, whose disappearance at the end of operations has been shown to be a predictor of long-term response [26].

## 4. Conclusions

In this work, we tried to provide a rapid review of different techniques we routinely use in CPA surgery. Neuronavigation is a useful tool that makes the CPA surgical approach faster and safer to perform. The different IONM techniques should be implemented in CPA surgery, especially in the cases of CPA tumors. Close communication and collaboration between the surgical team, the neurophysiologist, and the anesthesiologist are mandatory to obtain high-quality neuromonitoring, prevent neurologic injuries, and decrease the risk of misinterpretation of the neurophysiological changes.

## Figures and Tables

**Figure 1 brainsci-12-01167-f001:**
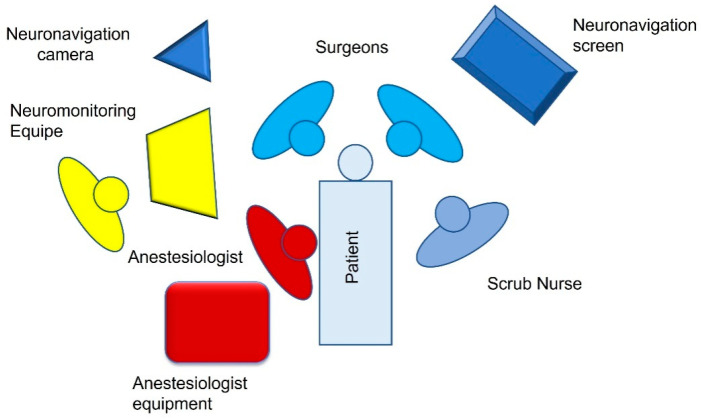
OR setting for left CPA surgery.

**Figure 2 brainsci-12-01167-f002:**
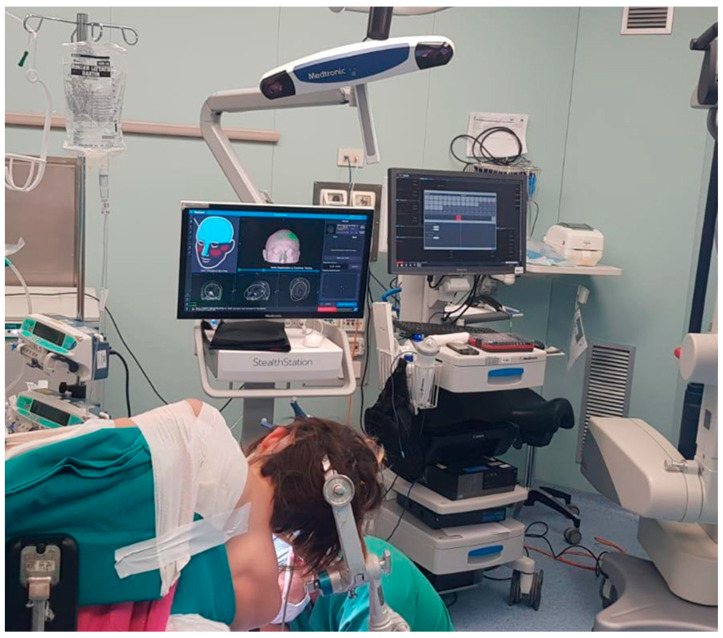
Park-bench position for neuronavigated assisted CPA surgery.

**Figure 3 brainsci-12-01167-f003:**
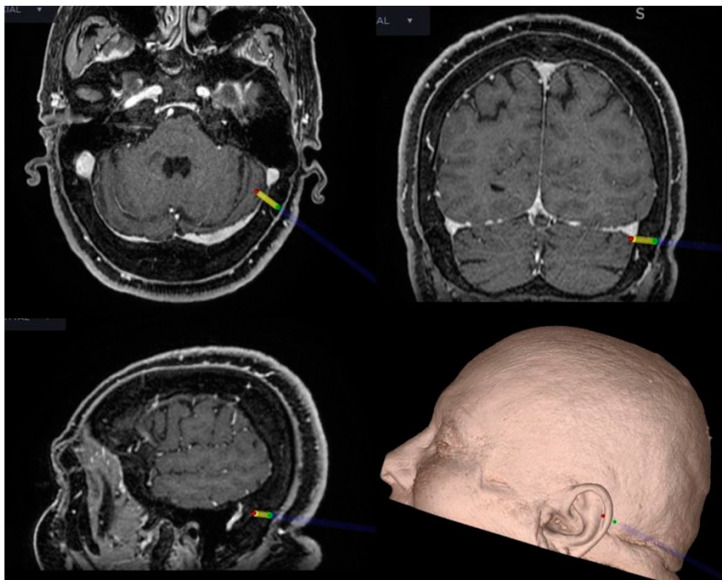
Intraoperative neuronavigation in CPA surgery. The key point for the craniotomy is set just behind and below the TSSJ.

**Figure 4 brainsci-12-01167-f004:**
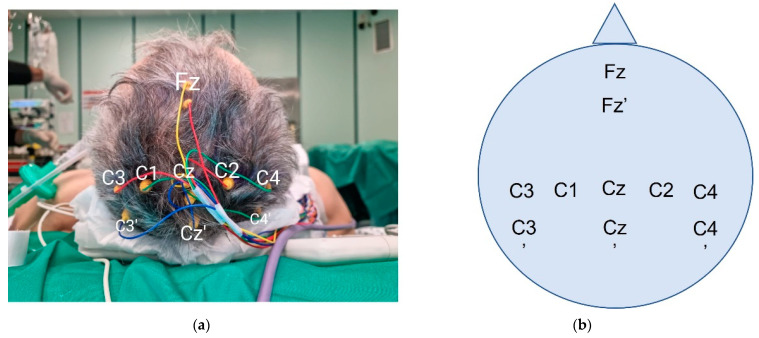
Intraoperative corkscrew scalp positioning (photograph (**a**) and schema (**b**)). C3 and C4 are both represented in this case. The choice is dictated by the affected side.

**Figure 5 brainsci-12-01167-f005:**
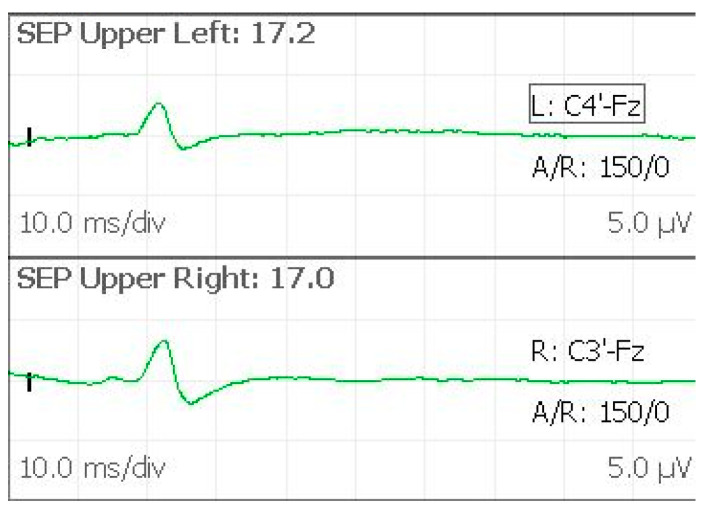
SEPs evoked from upper limbs, with the N20 peak symmetrically observed.

**Figure 6 brainsci-12-01167-f006:**
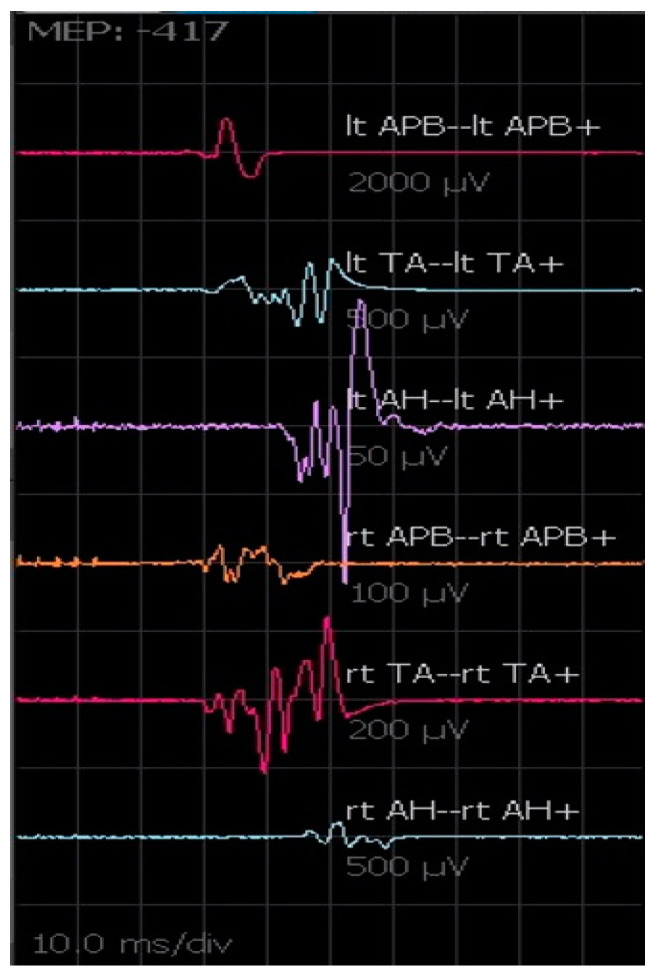
MEPs registered from the upper and lower limbs. The amplitude and morphology of the waves can change from one transcranial electrical stimulation to another.

**Figure 7 brainsci-12-01167-f007:**
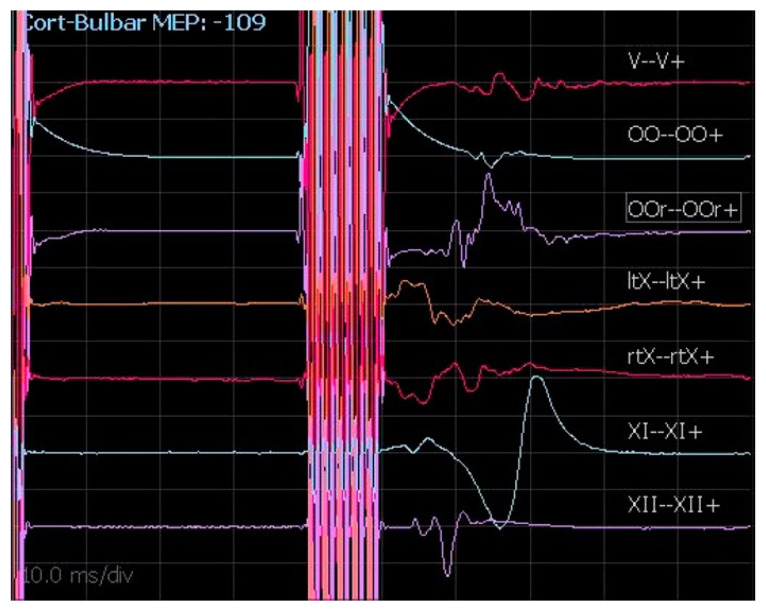
cMEPs registered from V, VII, X (left and right), XI, and XII cranial nerves.

**Figure 8 brainsci-12-01167-f008:**
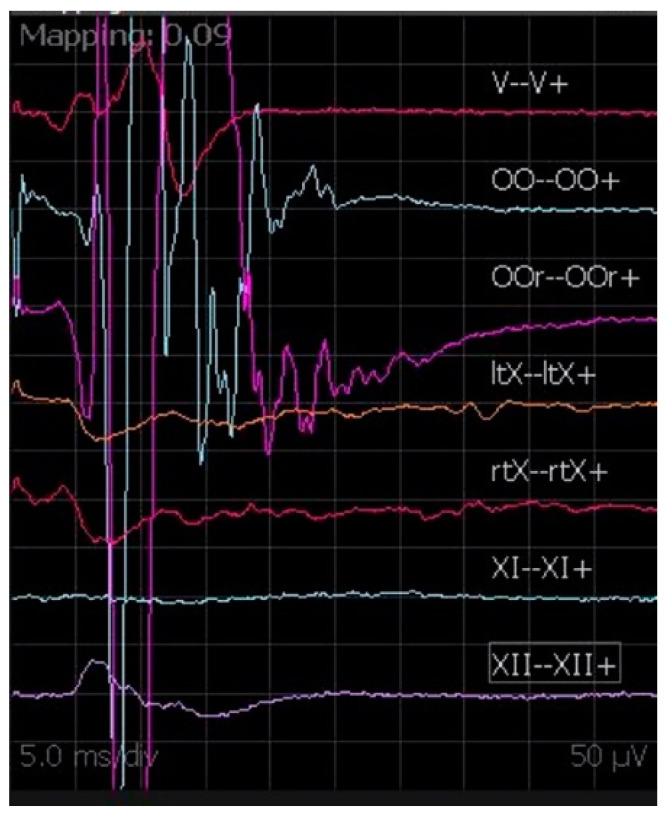
DES during vestibular schwannoma surgery, with muscular activation of OOr and OOc.

**Table 1 brainsci-12-01167-t001:** Technical parameters of stimulation and registration.

	Stimulation	Registration
SEPs	single pulse, 300–500 µs pulse duration 4.1 Hz; 10–25 mA	C3′–Fz (right stimulus) C4′–Fz (left stimulus)
MEPs	train of stimuli 5–7 pulse 500 µs pulse duration 250–500 Hz; 150–300 V	C1–C2 (anodal)
cMEPs	single pulse 350–500 µs; train of pulses 4–7 stimuli, 350–500 µs, 500–700 Hz, delivered at 40 ms from the first single pulse; 75–150 V	C3-Fz (right cranial nerves) C4-Fz (left cranial nerves)

**Table 2 brainsci-12-01167-t002:** Possible causes of SEPs variations.

Causes of SEPs Variation	
Technical	Electrode disconnection
	Equipment malfunction
	Local interference
Systemic	Hypoxia
	Hypotension
Surgical	Direct blunt trauma Excessive compression or traction Thermic damage Hypoperfusion
Anesthesiological	Propofol bolus Halogenated inhalational agents NO_2_

## Data Availability

Not applicable.

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
