# Peer review of "Tailored Approach and Multimodal Intraoperative Neuromonitoring in Cerebellopontine Angle Surgery"

_brainsci, 2022, doi:10.3390/brainsci12091167_

Round 1

Reviewer 1 Report

This is a quite reasonable review summarizing all methods which   constitute a contemporary neuromonitoring system used in modern neurosurgery. The cerebro-pontine angle surgery was indicated as a target for all these well orchestrated methods of  navigation and monitoring.

Formally there is nothing wrong with this manuscript; all the methods are described properly and professionally. Nevertheless my main concenrn is aimed at low originality of the content . The authors simply describe details peratining to to the application and interpretation of the subsequent methods but  this can be found in professiona handbooks of neurosurgery. As a matter of fact this article seems to me  more appropriate as a chapter of a professional handbook than as an original contribution to a scientific journal, especially such pertaining to general neuroscience rather than specifically to clinical neurosurgery. Therefore,  acknowledging essential merit of the manuscript, I must leave final decision  on this review  for consideration whether it really falls in the scope of the journal. Possibly a  neurosurgical journal with clinical scope would be a more appropriate vehicle for such contribution?

Minor remarks.

I wonder if the complete system of monitoring is really necessary in every case of CPA surgery. For example in Vth nerve decompressions for neuralgia  the anatomy is generally preserved   and the nerve so and so must be made free of any compression. In a majority of cases  all the procedure can be safely performed just with a neat preparation. Is therefore  a real need of employing all this costly ant time consuming system in every case of CPA surgery? A comment on this aspect could be added. 

Conclusion:

exploration rather than explosion?

Author Response

Response to Reviewers

Reviewer 1

I wonder if the complete system of monitoring is really necessary in every case of CPA surgery. For example in Vth nerve decompressions for neuralgia the anatomy is generally preserved   and the nerve so and so must be made free of any compression. In a majority of cases  all the procedure can be safely performed just with a neat preparation. Is therefore  a real need of employing all this costly ant time consuming system in every case of CPA surgery? A comment on this aspect could be added.

Thank you for your comment. We agree with you. We added the following sentence in the discussion section: “Obviously, due to the costs of these tools it is impossible to use all these techniques for every case. For example, in cases of microvascular decompression of trigeminal nerve, we routinely use the neuronavigation to plan the craniotomy but the intraoperative neuromonitoring is not mandatory due to the fact that the anatomy is usually preserved. On the other hand, we consider intraoperative neuromonitoring mandatory in cases of CPA tumors to preserve brainstem and cranial nerve function [16, 28]. Furthermore, intraoperative neuromonitoring can be very useful in cases of microvascular decompression of VII cranial nerve for hemifacial spasm to study the lateral spread responses whose disappearance at the end of operation has been shown to be a predictor of long-term response [25]

exploration rather than explosion?

Thank you. We fixed this mistake

Reviewer 2

First used at line 142, "N20" term should be explained in detail.

We better explained the “N20” meaning adding the following sentence: “We can identify the N20 analyzing peak morphology, latency and amplitude, (Figure 5). The N20 is a negative wave observed with a latency of about 20msec representing the activation of primary somatosensory cortex in the hand area”

At line 212, "OOc and OOr" terms are used. Please clarify.

We clarified in the text: orbicularis oculi (OOc) and orbicularis oris (OOr)

"Neuronavigation" section in "Operating Room Tools” is too short. That section should be enriched with figures including radiological images and patient positioning pictures.

We added the Figure 2 and the Figure 3 showing the patient’s position and the intraoperative radiological images

There are a few English language misusage in the manuscript. If not, English language editing should be performed preferably by a professional manuscript proofreading service.

English was carefully reviewed

Reviewer 3

The introduction is too short and should give both a historical and anatomical overview to inform the reader about the specific challenges that have been overcome by the neurosurgical community over the decades. It must be mentioned that already Cushing and Dandy debated this approach due to its challenges. Respective literature is for example cited in: Kaye, A. H., Briggs, R. J. S., & Morokoff, A. P. (2012). Acoustic neurinoma (vestibular schwannoma). Brain Tumors, 518–569. doi:10.1016/b978-0-443-06967-3.00028-4.

We significantly increased the introduction section citing the pertinent literature and adding the following paragraph: “The anatomy related to the retrosigmoid approach and the CPA has been extensively de-scribed in many papers and textbooks in the past years. It is worth to mention that in or-der to simplify the CPA approach, Rhoton proposed the “anatomical rule of three” divid-ing the CPA in three compartments according to the three pairs of cerebellar arteries aris-ing from the vertebrobasilar system[2]. He identified: 1) the “upper neurovascular com-plex” following the superior cerebellar artery and including the midbrain and upper pons, trigeminal nerve, trochlear nerve, superior petrosal vein and its tributaries, and superior cerebellum, 2) the “middle neurovascular complex” following the anterior inferior cere-bellar artery and including the middle pons, middle cerebellar peduncle, abducens, facial and vestibulocochlear nerves and 3) the “lower neurovascular complex” following the posterior inferior cerebellar artery and including the medulla, lower cranial nerves, and inferior cerebellum. Over the past years, the retrosigmoid craniotomy has evolved, initially from a bilateral suboccipital to a unilateral approach. Cushing reported a very impressive morbidity and mortality rate (about 80%) using the unilateral suboccipital approach and proposed a large bilateral suboccipital craniotomy in order to rapidly open the cisterna magna and obtain CSF release. Nonetheless, Dandy continued to promote the unilateral approach combining CSF release, cerebellar retraction and eventually amputation of a portion of cerebellum to gain access to CPA. In the recent years, the use of surgical micro-scope enabled the visualization of the critical neural and vascular structures significantly increasing the possibility of a CPA tumor radical resection with cranial nerve function preservation.”

There must be an extensive discussion of the presented setup in light of the existing literature. For example, how relevant is intraoperative monitoring (IOM) for outcome. This is for example discussed here: Arlt F, Kasper J, Winkler D, Jähne K, Fehrenbach MK, Meixensberger J, Sander C. Facial Nerve Function After Microsurgical Resection in Vestibular Schwannoma Under Neurophysiological Monitoring. Front Neurol. 2022 May 24;13:850326. doi: 10.3389/fneur.2022.850326. PMID: 35685739; PMCID: PMC9170892.

The discussion should comprise circa one page in the used template.

We added a discussion paragraph to our paper underlying the importance of IONM for the outcome in CPA surgery citing the pertinent literature

Reviewer 2 Report

Dear Authors,

Thank you for submitting your manuscript about multi-disciplinary and multi-modality approach for minimizing risks in cerebellopontine angle surgery. Study is written as a "Technical Note" format and gives highly valuable information regarding neuronavigation and neuromonitorization. As mentioned in the text, and also as I believe, neuronavigation is not only used for localizing the tumor or checking the extent of resection, it is also important for planning of every stage in operating room. "Brain Sciences" journal doesn’t accept technical notes, as far as I can understand, however the manuscript is highly attractive within the context and scope of the journal. The text
needs some minor revisions before consideration for publication:

1) First used at line 142, "N20" term should be explained in detail.

2) At line 212, "OOc and OOr" terms are used. Please clarify.

3) "Neuronavigation" section in "Operating Room Tools” is too short. That section should be enriched with figures including radiological images and patient positioning pictures.

4) There are a few English language misusage in the manuscript. If not, English language editing should be performed preferably by a professional manuscript proofreading service.

My opinion is that interest to the readers is high in the study.

Best regards.

Author Response

(The authors gave the same response as above.)

Reviewer 3 Report

The authors present their approach to the cerebello-pontine angle in this review. This is a common approach with a long history in neurosurgery. Therefore, the overall merit of the work is limited and improvements are required if publication should become feasible. 

The setup presentation is thorough and shows the sophisticated everyday design of the authors' institution. 

The introduction is too short and should give both a historical and anatomical overview to inform the reader about the specific challenges that have been overcome by the neurosurgical community over the decades. It must be mentioned that already Cushing and Dandy debated this approach due to its challenges. Respective literature is for example cited in: Kaye, A. H., Briggs, R. J. S., & Morokoff, A. P. (2012). Acoustic neurinoma (vestibular schwannoma). Brain Tumors, 518–569. doi:10.1016/b978-0-443-06967-3.00028-4.  

There must be an extensive discussion of the presented setup in light of the existing literature. For example, how relevant is intraoperative monitoring (IOM) for outcome. This is for example discussed here: Arlt F, Kasper J, Winkler D, Jähne K, Fehrenbach MK, Meixensberger J, Sander C. Facial Nerve Function After Microsurgical Resection in Vestibular Schwannoma Under Neurophysiological Monitoring. Front Neurol. 2022 May 24;13:850326. doi: 10.3389/fneur.2022.850326. PMID: 35685739; PMCID: PMC9170892. 

The discussion should comprise circa one page in the used template. 

Author Response

(The authors gave the same response as above.)

Round 2

Reviewer 3 Report

The authors made significant changes throughout their manuscript which is therefore now feasible for publication.